# Efficient Optimization for
# Sparse Gaussian Process Regression

**Yanshuai Cao**[1]   **Marcus A. Brubaker**[2]   **David J. Fleet**[1]   **Aaron Hertzmann**[1,3]

[1]Department of Computer Science    [2]TTI-Chicago    [3]Adobe Research
University of Toronto

## Abstract

We propose an efficient optimization algorithm for selecting a subset of training data to induce sparsity for Gaussian process regression. The algorithm estimates an inducing set and the hyperparameters using a single objective, either the marginal likelihood or a variational free energy. The space and time complexity are linear in training set size, and the algorithm can be applied to large regression problems on discrete or continuous domains. Empirical evaluation shows state-of-art performance in discrete cases and competitive results in the continuous case.

## 1   Introduction

Gaussian Process (GP) learning and inference are computationally prohibitive with large datasets, having time complexities $O(n^3)$ and $O(n^2)$, where $n$ is the number of training points. Sparsification algorithms exist that scale linearly in the training set size (see [10] for a review). They construct a low-rank approximation to the GP covariance matrix over the full dataset using a small set of *inducing points*. Some approaches select inducing points from training points [7, 8, 12, 13]. But these methods select the inducing points using ad hoc criteria; i.e., they use different objective functions to select inducing points and to optimize GP hyperparameters. More powerful sparsification methods [14, 15, 16] use a single objective function and allow inducing points to move freely over the input domain which are learned via gradient descent. This continuous relaxation is not feasible, however, if the input domain is discrete, or if the kernel function is not differentiable in the input variables. As a result, there are problems in myraid domains, like bio-informatics, linguistics and computer vision where current sparse GP regression methods are inapplicable or ineffective.

We introduce an efficient sparsification algorithm for GP regression. The method optimizes a single objective for joint selection of inducing points and GP hyperparameters. Notably, it optimizes either the marginal likelihood, or a variational free energy [15], exploiting the QR factorization of a partial Cholesky decomposition to efficiently approximate the covariance matrix. Because it chooses inducing points from the training data, it is applicable to problems on discrete or continuous input domains. To our knowledge, it is the first method for selecting discrete inducing points and hyperparameters that optimizes a single objective, with linear space and time complexity. It is shown to outperform other methods on discrete datasets from bio-informatics and computer vision. On continuous domains it is competitive with the Pseudo-point GP [14] (SPGP).

### 1.1   Previous Work

Efficient state-of-the-art sparsification methods are $O(m^2n)$ in time and $O(mn)$ in space for learning. They compute the predictive mean and variance in time $O(m)$ and $O(m^2)$. Methods based on continuous relaxation, when applicable, entail learning $O(md)$ continuous parameters, where $d$ is the input dimension. In the discrete case, combinatorial optimization is required to select the inducing points, and this is, in general, intractable. Existing discrete sparsification methods therefore use other criteria to greedily select inducing points [7, 8, 12, 13]. Although their criteria are justified, each in their own way (*e.g.*, [8, 12] take an information theoretic perspective), they are greedy and do not use the same objective to select inducing points and to estimate GP hyperparameters.

The variational formulation of Titsias [15] treats inducing points as variational parameters, and gives a unified objective for discrete and continuous inducing point models. In the continuous case, it uses gradient-based optimization to find inducing points and hyperparameters. In the discrete case, our method optimizes the same variational objective of Titsias [15], but is a significant improvement over greedy forward selection using the variational objective as selection criteria, or some other criteria. In particular, given the cost of evaluating the variational objective on all training points, Titsias [15] evaluates the objective function on a small random subset of candidates at each iteration, and then select the best element from the subset. This approximation is often slow to achieve good results, as we explain and demonstrate below in Section 4.1. The approach in [15] also uses greedy forward selection, which provides no way to refine the inducing set after hyperparameter optimization, except to discard all previous inducing points and restart selection. Hence, the objective is not guaranteed to decrease after each restart. By comparison, our formulation considers all candidates at each step, and revisiting previous selections is efficient, and guaranteed to decrease the objective or terminate.

Our low-rank decomposition is inspired by the *Cholesky with Side Information* (CSI) algorithm for kernel machines [1]. We extend that approach to GP regression. First, we alter the form of the low-rank matrix factorization in CSI to be suitable for GP regression with full-rank diagonal term in the covariance. Second, the CSI algorithm selects inducing points in a single greedy pass using an approximate objective. We propose an iterative optimization algorithm that swaps previously selected points with new candidates that are guaranteed to lower the objective. Finally, we perform inducing set selection jointly with gradient-based hyperparameter estimation instead of the grid search in CSI. Our algorithm selects inducing points in a principled fashion, optimizing the variational free energy or the log likelihood. It does so with time complexity $O(m^2 n)$, and in practice provides an improved quality-speed trade-off over other discrete selection methods.

## 2  Sparse GP Regression

Let $y \in \mathbb{R}$ be the noisy output of a function, $f$, of input $\mathbf{x}$. Let $X = \{\mathbf{x}_i\}_{i=1}^n$ denote $n$ training inputs, each belonging to input space $\mathcal{D}$, which is not necessarily Euclidean. Let $\mathbf{y} \in \mathbb{R}^n$ denote the corresponding vector of training outputs. Under a full zero-mean GP, with the covariance function

$$\mathbb{E}[y_i y_j] = \kappa(\mathbf{x}_i, \mathbf{x}_j) + \sigma^2 1[i = j] \,, \tag{1}$$

where $\kappa$ is the kernel function, $1[\cdot]$ is the usual indicator function, and $\sigma^2$ is the variance of the observation noise, the predictive distribution over the output $f_\star$ at a test point $\mathbf{x}_\star$ is normally distributed. The mean and variance of the predictive distribution can be expressed as

$$\mu_\star = \boldsymbol{\kappa}(\mathbf{x}_\star)^\mathsf{T} \left( K + \sigma^2 I_n \right)^{-1} \mathbf{y}$$
$$v_\star^2 = \kappa(\mathbf{x}_\star, \mathbf{x}_\star) - \boldsymbol{\kappa}(\mathbf{x}_\star)^\mathsf{T} \left( K + \sigma^2 I_n \right)^{-1} \boldsymbol{\kappa}(\mathbf{x}_\star)$$

where $I_n$ is the $n \times n$ identity matrix, $K$ is the kernel matrix whose $ij$th element is $\kappa(\mathbf{x}_i, \mathbf{x}_j)$, and $\boldsymbol{\kappa}(\mathbf{x}_\star)$ is the column vector whose $i$th element is $\kappa(\mathbf{x}_\star, \mathbf{x}_i)$.

The hyperparameters of a GP, denoted $\boldsymbol{\theta}$, comprise the parameters of the kernel function, and the noise variance $\sigma^2$. The natural objective for learning $\boldsymbol{\theta}$ is the negative marginal log likelihood (NMLL) of the training data, $-\log\left(P(\mathbf{y}|X, \boldsymbol{\theta})\right)$, given up to a constant by

$$E_{full}(\boldsymbol{\theta}) = \left( \mathbf{y}^\top \left( K + \sigma^2 I_n \right)^{-1} \mathbf{y} + \log|K + \sigma^2 I_n| \right) / 2 \,. \tag{2}$$

The computational bottleneck lies in the $O(n^2)$ storage and $O(n^3)$ inversion of the full covariance matrix, $K + \sigma^2 I_n$. To lower this cost with a sparse approximation, Csató and Opper [5] and Seeger *et al.* [12] proposed the Projected Process (PP) model, wherein a set of $m$ inducing points are used to construct a low-rank approximation of the kernel matrix. In the discrete case, where the inducing points are a subset of the training data, with indices $\mathcal{I} \subset \{1, 2, ..., n\}$, this approach amounts to replacing the kernel matrix $K$ with the following Nyström approximation [11]:

$$K \simeq \hat{K} = K[:, \mathcal{I}] \, K[\mathcal{I}, \mathcal{I}]^{-1} \, K[\mathcal{I}, :] \tag{3}$$

where $K[:, \mathcal{I}]$ denotes the sub-matrix of $K$ comprising columns indexed by $\mathcal{I}$, and $K[\mathcal{I}, \mathcal{I}]$ is the sub-matrix of $K$ comprising rows and columns indexed by $\mathcal{I}$. We assume the rank of $K$ is $m$ or higher so we can always find such rank-$m$ approximations. The PP NMLL is then algebraically equivalent to replacing $K$ with $\hat{K}$ in Eq. (2), *i.e.*,

$$E(\boldsymbol{\theta}, \mathcal{I}) = \left( E^D(\boldsymbol{\theta}, \mathcal{I}) + E^C(\boldsymbol{\theta}, \mathcal{I}) \right) / 2 \,, \tag{4}$$

with data term $E^D(\boldsymbol{\theta}, \mathcal{I}) = \mathbf{y}^\top (\hat{K} + \sigma^2 I_n)^{-1} \mathbf{y}$, and model complexity $E^C(\boldsymbol{\theta}, \mathcal{I}) = \log |\hat{K} + \sigma^2 I_n|$.

The computational cost reduction from $O(n^3)$ to $O(m^2 n)$ associated with the new likelihood is achieved by applying the Woodbury inversion identity to $E^D(\boldsymbol{\theta}, \mathcal{I})$ and $E^C(\boldsymbol{\theta}, \mathcal{I})$. The objective in (4) can be viewed as an approximate log likelihood for the full GP model, or as the exact log likelihood for an approximate model, called the Deterministically Trained Conditional [10].

The same PP model can also be obtained by a variational argument, as in [15], for which the variational free energy objective can be shown to be Eq. (4) plus one extra term; *i.e.*,

$$F(\boldsymbol{\theta}, \mathcal{I}) \;=\; \left( E^D(\boldsymbol{\theta}, \mathcal{I}) + E^C(\boldsymbol{\theta}, \mathcal{I}) + E^V(\boldsymbol{\theta}, \mathcal{I}) \right) / 2 \,, \tag{5}$$

where $E^V(\boldsymbol{\theta}, \mathcal{I}) = \sigma^{-2} \operatorname{tr}(K - \hat{K})$ arises from the variational formulation. It effectively regularizes the trace norm of the approximation residual of the covariance matrix. The kernel machine of [1] also uses a regularizer of the form $\lambda \operatorname{tr}(K - \hat{K})$, however $\lambda$ is a free parameter that is set manually.

## 3 Efficient optimization

We now outline our algorithm for optimizing the variational free energy (5) to select the inducing set $\mathcal{I}$ and the hyperparameters $\boldsymbol{\theta}$. (The negative log-likelihood (4) is similarly minimized by simply discarding the $E^V$ term.) The algorithm is a form of hybrid coordinate descent that alternates between discrete optimization of inducing points, and continuous optimization of the hyperparameters. We first describe the algorithm to select inducing points, and then discuss continuous hyperparameter optimization and termination criteria in Sec. 3.4.

Finding the optimal inducing set is a combinatorial problem; global optimization is intractable. Instead, the inducing set is initialized to a random subset of the training data, which is then refined by a fixed number of swap updates at each iteration.[1] In a single swap update, a randomly chosen inducing point is considered for replacement. If swapping does not improve the objective, then the original point is retained. There are $n - m$ potential replacements for each each swap update; the key is to efficiently determine which will maximally improve the objective. With the techniques described below, the computation time required to approximately evaluate all possible candidates and swap an inducing point is $O(mn)$. Swapping all inducing points once takes $O(m^2 n)$ time.

### 3.1 Factored representation

To support efficient evaluation of the objective and swapping, we use a factored representation of the kernel matrix. Given an inducing set $\mathcal{I}$ of $k$ points, for any $k \leq m$, the low-rank Nyström approximation to the kernel matrix (Eq. 3) can be expressed in terms of a partial Cholesky factorization:

$$\hat{K} \;=\; K[:, \mathcal{I}]\, K[\mathcal{I}, \mathcal{I}]^{-1}\, K[\mathcal{I}, :] \;=\; L(\mathcal{I}) L(\mathcal{I})^\top \,, \tag{6}$$

where $L(\mathcal{I}) \in \mathbb{R}^{n \times k}$ is, up to permutation of rows, lower trapezoidal matrix (*i.e.*, has a $k \times k$ top lower triangular block, again up to row permutation). The derivation of Eq. 6 follows from Proposition 1 in [1], and the fact that, given the ordered sequence of pivots $\mathcal{I}$, the partial Cholesky factorization is unique.

Using this factorization and the Woodbury identities (dropping the dependence on $\boldsymbol{\theta}$ and $\mathcal{I}$ for clarity), the terms of the negative marginal log-likelihood (4) and variational free energy (5) become

$$E^D \;=\; \sigma^{-2} \left( \mathbf{y}^\top \mathbf{y} - \mathbf{y}^\top L \left( L^\top L + \sigma^2 I \right)^{-1} L^\top \mathbf{y} \right) \tag{7}$$

$$E^C \;=\; \log \left( (\sigma^2)^{n-k} |L^\top L + \sigma^2 I| \right) \tag{8}$$

$$E^V \;=\; \sigma^{-2} (\operatorname{tr}(K) - \operatorname{tr}(L^\top L)) \tag{9}$$

We can further simplify the data term by augmenting the factor matrix as $\widetilde{L} = [L^\top, \ \sigma I_k]^\top$, where $I_k$ is the $k \times k$ identity matrix, and $\widetilde{\mathbf{y}} = [\mathbf{y}^\top, \mathbf{0}_k^\top]^\top$ is the $\mathbf{y}$ vector with $k$ zeroes appended:

$$E^D = \sigma^{-2} \left( \mathbf{y}^\top \mathbf{y} - \widetilde{\mathbf{y}}^\top \widetilde{L} (\widetilde{L}^\top \widetilde{L})^{-1} \widetilde{L}^\top \widetilde{\mathbf{y}} \right) \tag{10}$$

Now, let $\widetilde{L} = QR$ be a QR factorization of $\widetilde{L}$, where $Q \in \mathbb{R}^{(n+k) \times k}$ has orthogonal columns and $R \in \mathbb{R}^{k \times k}$ is invertible. The first two terms in the objective simplify further to

$$E^D = \sigma^{-2} \left( \|\mathbf{y}\|^2 - \|Q^\top \widetilde{\mathbf{y}}\|^2 \right) \tag{11}$$

$$E^C = (n - k) \log(\sigma^2) + 2 \log |R| . \tag{12}$$

## 3.2 Factorization update

Here we present the mechanics of the swap update algorithm, see [3] for pseudo-code. Suppose we wish to swap inducing point $i$ with candidate point $j$ in $\mathcal{I}_m$, the inducing set of size $m$. We first modify the factor matrices in order to remove point $i$ from $\mathcal{I}_m$, *i.e.* to downdate the factors. Then we update all the key terms using one step of Cholesky and QR factorization with the new point $j$.

Downdating to remove inducing point $i$ requires that we shift the corresponding columns/rows in the factorization to the right-most columns of $\widetilde{L}$, $Q$, $R$ and to the last row of $R$. We can then simply discard these last columns and rows, and modify related quantities. When permuting the order of the inducing points, the underlying GP model is invariant, but the matrices in the factored representation are not. If needed, any two points in $\mathcal{I}_m$, can be permuted, and the Cholesky or QR factors can be updated in time $O(mn)$. This is done with the efficient pivot permutation presented in the Appendix of [1], with minor modifications to account for the augmented form of $\widetilde{L}$. In this way, downdating and removing $i$ take $O(mn)$ time, as does the updating with point $j$.

After downdating, we have factors $\widetilde{L}_{m-1}, Q_{m-1}, R_{m-1}$, and inducing set $\mathcal{I}_{m-1}$. To add $j$ to $\mathcal{I}_{m-1}$, and update the factors to rank $m$, one step of Cholesky factorization is performed with point $j$, for which, ideally, the new column to append to $\widetilde{L}$ is formed as

$$\boldsymbol{\ell}_m = (K - \hat{K}_{m-1})[:, j] \left/ \sqrt{(K - \hat{K}_{m-1})[j, j]} \right. \tag{13}$$

where $\hat{K}_{m-1} = L_{m-1} L_{m-1}^\top$. Then, we set $\widetilde{L}_m = [\widetilde{L}_{m-1} \; \tilde{\boldsymbol{\ell}}_m]$, where $\tilde{\boldsymbol{\ell}}_m$ is just $\boldsymbol{\ell}_m$ augmented with $\sigma \mathbf{e}_m = [0, 0, ..., \sigma, ..., 0, 0]^\top$. The final updates are $Q_m = [Q_{m-1} \; \mathbf{q}_m]$, where $\mathbf{q}_m$ is given by Gram-Schmidt orthogonalization, $\mathbf{q}_m = ((I - Q_{m-1} Q_{m-1}^\top) \tilde{\boldsymbol{\ell}}_m) / \|(I - Q_{m-1} Q_{m-1}^\top) \tilde{\boldsymbol{\ell}}_m\|$, and $R_m$ is updated from $R_{m-1}$ so that $\widetilde{L}_m = Q_m R_m$.

## 3.3 Evaluating candidates

Next we show how to select candidates for inclusion in the inducing set. We first derive the exact change in the objective due to adding an element to $\mathcal{I}_{m-1}$. Later we will provide an approximation to this objective change that can be computed efficiently.

Given an inducing set $\mathcal{I}_{m-1}$, and matrices $\widetilde{L}_{m-1}, Q_{m-1}$, and $R_{m-1}$, we wish to evaluate the change in Eq. 5 for $\mathcal{I}_m = \mathcal{I}_{m-1} \cup j$. That is, $\Delta F \equiv F(\boldsymbol{\theta}, \mathcal{I}_{m-1}) - F(\boldsymbol{\theta}, \mathcal{I}_m) = (\Delta E^D + \Delta E^C + \Delta E^V)/2$, where, based on the mechanics of the incremental updates above, one can show that

$$\Delta E^D = \sigma^{-2} (\widetilde{\mathbf{y}}^\top \left( I - Q_{m-1} Q_{m-1}^\top \right) \tilde{\boldsymbol{\ell}}_m)^2 \left/ \| \left( I - Q_{m-1} Q_{m-1}^\top \right) \tilde{\boldsymbol{\ell}}_m \|^2 \right. \tag{14}$$

$$\Delta E^C = \log \left( \sigma^2 \right) - \log \| (I - Q_{m-1} Q_{m-1}^\top) \tilde{\boldsymbol{\ell}}_m \|^2 \tag{15}$$

$$\Delta E^V = \sigma^{-2} \|\boldsymbol{\ell}_m\|^2 \tag{16}$$

This gives the exact decrease in the objective function after adding point $j$. For a single point this evaluation is $O(mn)$, so to evaluate all $n - m$ points would be $O(mn^2)$.

### 3.3.1 Fast approximate cost reduction

While $O(mn^2)$ is prohibitive, computing the exact change is not required. Rather, we only need a ranking of the best few candidates. Thus, instead of evaluating the change in the objective exactly, we use an efficient approximation based on a small number, $z$, of training points which provide information about the residual between the current low-rank covariance matrix (based on inducing points) and the full covariance matrix. After this approximation proposes a candidate, we use the actual objective to decide whether to include it. The techniques below reduce the complexity of evaluating all $n - m$ candidates to $O(zn)$.

To compute the change in objective for one candidate, we need the new column of the updated Cholesky factorization, $\boldsymbol{\ell}_m$. In Eq. (13) this vector is a (normalized) column of the residual

$K - \hat{K}_{m-1}$ between the full kernel matrix and the Nyström approximation. Now consider the full Cholesky decomposition of $K = L^* L^{*\top}$ where $L^* = [L_{m-1}, L(\mathcal{J}_{m-1})]$ is constructed with $\mathcal{I}_{m-1}$ as the first pivots and $\mathcal{J}_{m-1} = \{1,...,n\}\backslash\mathcal{I}_{m-1}$ as the remaining pivots, so the residual becomes $K - \hat{K}_{m-1} = L(\mathcal{J}_{m-1})L(\mathcal{J}_{m-1})^\top$. We approximate $L(\mathcal{J}_{m-1})$ by a rank $z \ll n$ matrix, $L_z$, by taking $z$ points from $\mathcal{J}_{m-1}$ and performing a partial Cholesky factorization of $K - \hat{K}_{m-1}$ using these pivots. The residual approximation becomes $K - \hat{K}_{m-1} \approx L_z L_z^\top$, and thus $\ell_m \approx (L_z L_z^\top)[:,j] / \sqrt{(L_z L_z^\top)[j,j]}$. The pivots used to construct $L_z$ are called *information pivots*; their selection is discussed in Sec. 3.3.2.

The approximations to $\Delta E_k^D$, $\Delta E_k^C$ and $\Delta E_k^V$, Eqs. (14)-(16), for all candidate points, involve the following terms: $\operatorname{diag}(L_z L_z^\top L_z L_z^\top)$, $\mathbf{y}^\top L_z L_z^\top$, and $(Q_{k-1}[1:n,:])^\top L_z L_z^\top$. The first term can be computed in time $O(z^2 n)$, and the other two in $O(zmn)$ with careful ordering of matrix multiplications.[2] Computing $L_z$ costs $O(z^2 n)$, but can be avoided since information pivots change by at most one when an information pivots is added to the inducing set and needs to be replaced. The techniques in Sec. 3.2 bring the associated update cost to $O(zn)$ by updating $L_z$ rather than recomputing it. These $z$ information pivots are equivalent to the "look-ahead" steps of Bach and Jordan's CSI algorithm, but as described in Sec. 3.3.2, there is a more effective way to select them.

### 3.3.2 Ensuring a good approximation

Selection of the information pivots determines the approximate objective, and hence the candidate proposal. To ensure a good approximation, the CSI algorithm [1] greedily selects points to find an approximation of the residual $K - \hat{K}_{m-1}$ in Eq. (13) that is optimal in terms of a bound of the trace norm. The goal, however, is to approximate Eqs. (14)-(16). By analyzing the role of the residual matrix, we see that the information pivots provide a low-rank approximation to the orthogonal complement of the space spanned by current inducing set. With a fixed set of information pivots, parts of that subspace may never be captured. This suggests that we might occasionally update the entire set of information pivots. Although information pivots are changed when one is moved into the inducing set, we find empirically that this is not insufficient. Instead, at regular intervals we replace the entire set of information pivots by random selection. We find this works better than optimizing the information pivots as in [1].

Figure 1 compares the exact and approximate cost reduction for candidate inducing points (left), and their respective rankings (right). The approximation is shown to work well. It is also robust to changes in the number of information pivots and the frequency of updates. When bad candidates are proposed, they are rejected after evaluating the change in the true objective. We find that rejection rates are typically low during early iterations ($< 20\%$), but increase as optimization nears convergence (to $30\%$ or $40\%$). Rejection rates also increase for sparser models, where each inducing point plays a more critical role and is harder to replace.

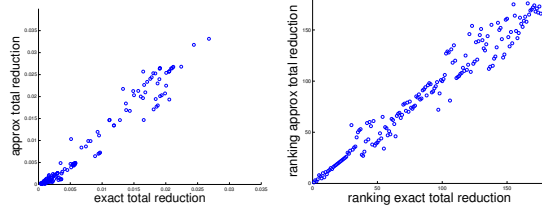

Figure 1: Exact vs approximate costs, based on the 1D example of Sec. 4, with $z = 10$, $n = 200$.

### 3.4 Hybrid optimization

The overall hybrid optimization procedure performs block coordinate descent in the inducing points and the continuous hyperparameters. It alternates between discrete and continuous phases until improvement in the objective is below a threshold or the computational time budget is exhausted.

In the discrete phase, inducing points are considered for swapping with the hyper-parameters fixed. With the factorization and efficient candidate evaluation above, swapping an inducing point $i \in \mathcal{I}_m$ proceeds as follows: (I) down-date the factorization matrices as in Sec. 3.2 to remove $i$; (II) compute the true objective function value $F_{m-1}$ over the down-dated model with $\mathcal{I}_m \backslash \{i\}$, using (11), (12) and (9); (III) select a replacement candidate using the fast approximate cost change from Sec. 3.3.1; (IV) evaluate the exact objective change, using (14), (15), and (16); (V) add the exact change to the true objective $F_{m-1}$ to get the objective value with the new candidate. If this improves, we include

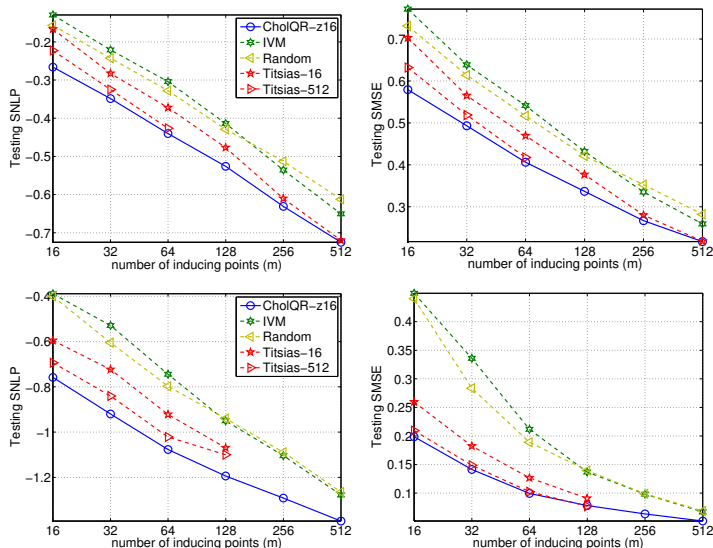

Figure 2: Test performance on discrete datasets. **(top row)** BindingDB, values at each marker is the average of 150 runs (50-fold random train/test splits times 3 random initialization); **(bottom row)** HoG dataset, each marker is the average of 10 randomly initialized runs.

the candidate in $\mathcal{I}$ and update the matrices as in Sec. 3.2. Otherwise it is rejected and we revert to the factorization with $i$; (VI) if needed, update the information pivots as in Secs. 3.3.1 and 3.3.2.

After each discrete optimization step we fix the inducing set $\mathcal{I}$ and optimize the hyperparameters using non-linear conjugate gradients (CG). The equivalence in (6) allows us to compute the gradient with respect to the hyperparameters analytically using the Nyström form. In practice, because we alternate each phase for many training epochs, attempting to swap every inducing point in each epoch is unnecessary, just as there is no need to run hyperparameter optimization until convergence. As long as all inducing set points are eventually considered we find that optimized models can achieve similar performance with shorter learning times.

## 4 Experiments and analysis

For the experiments that follow we jointly learn inducing points and hyperparameters, a more challenging task than learning inducing points with known hyperparameters [12, 14]. For all but the 1D example, the number of inducing points swapped per epoch is $min(60, m)$. The maximum number of function evaluations per epoch in CG hyperparameter optimization is $min(20, max(15, 2d))$, where $d$ is the number of continuous hyperparameters. Empirically we find the algorithm is robust to changes in these limits. We use two performance measures, (a) standardized mean square error (SMSE), $\frac{1}{N}\Sigma_{t=1}^{N}(\hat{y}_t - y_t)^2/\hat{\sigma}_*^2$, where $\hat{\sigma}_*^2$ is the sample variance of test outputs $\{y_t\}$, and (2) standardized negative log probability (SNLP) defined in [11].

### 4.1 Discrete input domain

We first show results on two discrete datasets with kernels that are not differentiable in the input variable $x$. Because continuous relaxation methods are not applicable, we compare to discrete selection methods, namely, random selection as baseline (Random), greedy subset-optimal selection of Titsias [15] with either 16 or 512 candidates (Titsias-16 and Titsias-512), and Informative Vector Machine [8] (IVM). For learning continuous hyperparameters, each method optimizes the same objective using non-linear CG. Care is taken to ensure consist initialization and termination criteria [3]. For our algorithm we use $z = 16$ information pivots with random selection (CholQR-z16). Later, we show how variants of our algorithm trade-off speed and performance. Additionally, we also compare to least-square kernel regression using CSI (in Fig. 3(c)).

The first discrete dataset, from `bindingdb.org`, concerns the prediction of binding affinity for a target (Thrombin), from the 2D chemical structure of small molecules (represented as graphs). We do 50-fold random splits to 3660 training points and 192 test points for repeated runs. We use a compound kernel, comprising 14 different graph kernels, and 15 continuous hyperparameters (one

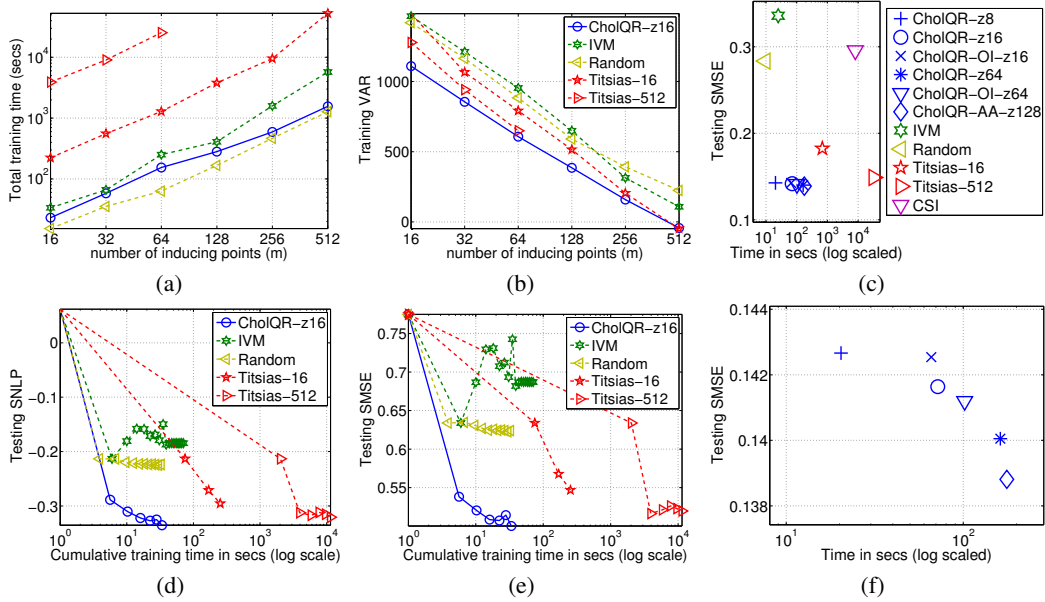

Figure 3: Training time versus test performance on discrete datasets. **(a)** the average BindingDB training time; **(b)** the average BindingDB objective function value at convergence; **(d)** and **(e)** show test scores versus training time with $m = 32$ for a single run; **(c)** shows the trade-off between training time and testing SMSE on the HoG dataset with $m = 32$, for various methods including multiple variants of CholQR and CSI; **(f)** a zoomed-in version of (c) comparing the variants of CholQR.

noise variance and 14 data variances). In the second task, from [2], the task is to predict 3D human joint position from histograms of HoG image features [6]. Training and test sets have 4819 and 4811 data points. Because our goal is the general purpose sparsification method for GP regression, we make no attempt at the more difficult problem of modelling the multivariate output structure in the regression as in [2]. Instead, we predict the vertical position of joints independently, using a histogram intersection kernel [9], having four hyperparameters: one noise variance, and three data variances corresponding to the kernel evaluated over the HoG from each of three cameras. We select and show result on the representative left wrist here (see [3] for others joints, and more details about the datasets and kernels used).

The results in Fig. 2 and 3 show that CholQR-z16 outperforms the baseline methods in terms of test-time predictive power with significantly lower training time. Titsias-16 and Titsias-512 shows similar test performance, but they are two to four orders of magnitude slower than CholQR-z16 (see Figs. 3(d) and 3(e)). Indeed, Fig. 3(a) shows that the training time for CholQR-z16 is comparable to IVM and Random selection, but with much better performance. The poor performance of Random selection highlights the importance of selecting good inducing points, as no amount of hyperparameter optimization can correct for poor inducing points. Fig. 3(a) also shows IVM to be somewhat slower due to the increased number of iterations needed, even though per epoch, IVM is faster than CholQR. When stopped earlier, IVM test performance further degrades.

Finally, Fig. 3(c) and 3(f) show the trade-off between the test SMSE and training time for variants of CholQR, with baselines and CSI kernel regression [1]. For CholQR we consider different numbers of information pivots (denoted z8, z16, z64 and z128), and different strategies for their selection including random selection, optimization as in [1] (denote OI) and adaptively growing the information pivot set (denoted AA, see [3] for details). These variants of CholQR trade-off speed and performance (3(f)), all significantly outperform the other methods (3(c)); CSI, which uses grid search to select hyper-parameters, is slow and exhibits higher SMSE.

## 4.2 Continuous input domain

Although CholQR was developed for discrete input domains, it can be competitive on continuous domains. To that end, we compare to SPGP [14] and IVM [8], using RBF kernels with one length-scale parameter per input dimension; $\kappa(\mathbf{x}_i, \mathbf{x}_j) = c \exp(-0.5 \sum_{t=1}^{d} b_t (\mathbf{x}_i^{(t)} - \mathbf{x}_j^{(t)})^2)$. We show results from both the PP log likelihood and variational objectives, suffixed by *MLE* and *VAR*.

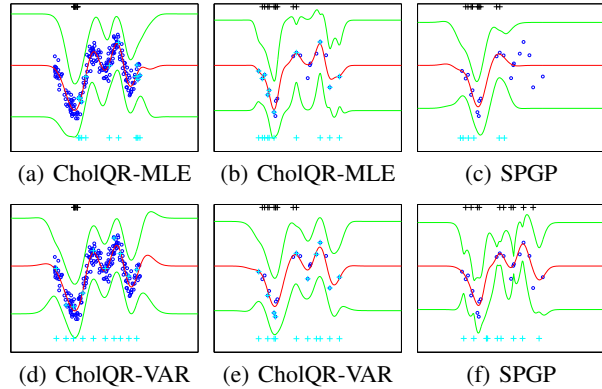

| (a) CholQR-MLE | (b) CholQR-MLE | (c) SPGP |
|---|---|---|

| (d) CholQR-VAR | (e) CholQR-VAR | (f) SPGP |
|---|---|---|

Figure 4: Snelson's 1D example: prediction mean (red curves); one standard deviation in prediction uncertainty (green curves); inducing point initialization (black points at top of each figure); learned inducing point locations (the cyan points at the bottom, also overlaid on data for CholQR).

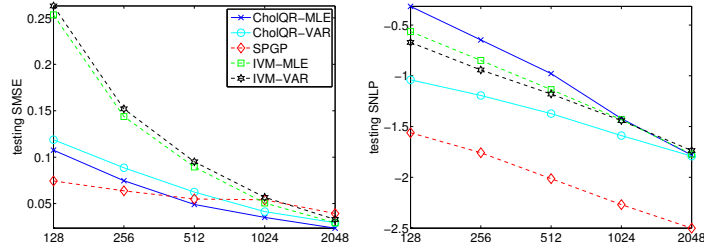

Figure 5: Test scores on KIN40K as function of number of inducing points: for each number of inducing points the value plotted is averaged over 10 runs from 10 different (shared) initializations.

We use the 1D toy dataset of [14] to show how the PP likelihood with gradient-based optimization of inducing points is easily trapped in local minima. Fig. 4(a) and 4(d) show that for this dataset our algorithm does not get trapped when initialization is poor (as in Fig. 1c of [14]). To simulate the sparsity of data in high-dimensional problems we also down-sample the dataset to 20 points (every 10th point). Here CholQR out-performs SPGP (see Fig. 4(b), 4(e), and 4(c)). By comparison, Fig. 4(f) shows SPGP learned with a more uniform initial distribution of inducing points avoids this local optima and achieves a better negative log likelihood of $11.34$ compared to $14.54$ in Fig. 4(c).

Finally, we compare CholQR to SPGP [14] and IVM [8] on a large dataset. *KIN40K* concerns nonlinear forward kinematic prediction. It has 8D real-valued inputs and scalar outputs, with 10K training and 30K test points. We perform linear de-trending and re-scaling as pre-processing. For SPGP we use the implementation of [14]. Fig. 5 shows that CholQR-VAR outperforms IVM in terms of SMSE and SNLP. Both CholQR-VAR and CholQR-MLE outperform SPGP in terms of SMSE on KIN40K with large $m$, but SPGP exhibits better SNLP. This disparity between the SMSE and SNLP measures for CholQR-MLE is consistent with findings about the PP likelihood in [15]. Recently, Chalupka *et al.* [4] introduced an empirical evaluation framework for approximate GP methods, and showed that subset of data (SoD) often compares favorably to more sophisticated sparse GP methods. Our preliminary experiments using this framework suggest that CholQR outperforms SPGP in speed and predictive scores; and compared to SoD, CholQR is slower during training, but proportionally faster during testing since CholQR finds a much sparser model to achieve the same predictive scores. In future work, we will report results on the complete suit of benchmark tests.

## 5 Conclusion

We describe an algorithm for selecting inducing points for Gaussian Process sparsification. It optimizes principled objective functions, and is applicable to discrete domains and non-differentiable kernels. On such problems it is shown to be as good as or better than competing methods and, for methods whose predictive behavior is similar, our method is several orders of magnitude faster. On continuous domains the method is competitive. Extension to the SPGP form of covariance approximation would be interesting future research.

## Footnotes

[1]The inducing set can be incrementally constructed, as in [1], however we found no benefit to this.

[2]Both can be further reduced to $O(zn)$ by appropriate caching during the updates of $Q,R$ and $\widetilde{L}$, and $L_z$

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
