[Supplementary Material]

# Efficient Optimization for Sparse Gaussian Process Regression: Supplementary Material

**Yanshuai Cao[1], Marcus A. Brubaker[2], David J. Fleet[1], Aaron Hertzmann[1,3]**

[1]Department of Computer Science    [2]TTI-Chicago    [3]Adobe Research
University of Toronto

This document is the supplementary material for the paper titled *Efficient Optimization for Sparse Gaussian Process Regression*.

## 1 Incremental Cholesky QR factorization

The following algorithm is essentially the incremental partial Cholesky factorization with column pivoting and the classical Graham-Schmidt factorization combined together.

$K$ is the rank $n$ full covariance matrix to be factorized, and $K$ does not need to precomputed (taking up $O(n^2)$ storage), but just need to return its diagonal and specific column when queried (a function handle for example). If $\sigma$ is supplied, the algorithm below operates with an additional twist allowing the augmentation trick introduced in Sec. 3 of the paper, in which case the matrix $L$ in the algorithm is the augmented version $\widetilde{L}$ mentioned in the paper; and $L[1:n,:]$ is the un-augmented portion. $Q$,$R$ are the QR factorization of $L$. The procedure also returns two vectors, $\mathbf{p}$ and $\mathbf{d}$. $\mathbf{p}$ is a permutation of $(1, 2, \ldots, n)$, and $\mathbf{d}$ stores the diagonal values of the residual matrix between the full $K$ and current partial Cholesky factorization. In our application to kernel covariance matrix, $\mathbf{d}$ is also the point-wise variance that is not yet explained by (the factorization using) existing inducing points. See post-conditions after the algorithm for formal relationships among various quantities.

For ease of description, explicit row pivoting is not performed (consistent with the description in section 3.1 of the paper). Instead, the ordering of rows of $L[1:n,:]$ always stays in the original order of data points $(1, 2, \ldots, n)$, and we use $\mathbf{p}$ to keep track the permutation, and index into the rows of $L[1:n,:]$. The columns are pivoted explicitly during the algorithm. In practical implementation however, we find the equivalent version with explicit row pivoting is slightly faster due to better memory/cache locality.

Assuming that the inducing set $\mathcal{I}_m = [i_1, \ldots, i_k, \ldots, i_m]$ is known, to build the factors:

> **procedure** CHOLQR_MSTEP($\mathcal{I}_m$, n, K, $\sigma$)
>     $\mathbf{p} \leftarrow [1, 2, \ldots, n]$
>     $\mathbf{d} \leftarrow diag(K)$
>     **if** $\sigma$ is given **then** # Need to do the augmentation
>         $L \leftarrow zeros(n + m, m)$
>         $Q \leftarrow zeros(n + m, m)$
>     **else**
>         $L \leftarrow zeros(n, m)$
>         $Q \leftarrow zeros(n, m)$
>     **end if**
>     $R \leftarrow zeros(m, m)$
>     **for** $k = 1 \rightarrow m$ **do**
>         $t \leftarrow$ position of $\mathcal{I}_m[k]$ in $\mathbf{p}$
>         $\mathbf{p}, L, Q, R, \mathbf{d} \leftarrow$ CholQR_OneStep($t, k, n, K, \mathbf{p}, L, Q, R, \mathbf{d}, \sigma$)
>     **end for**
>     **return** p, L, Q, R, d

**end procedure**
**procedure** CHOLQR_ONESTEP(t, k, n, K, **p**, L, Q, R, **d**, $\sigma$)
    $\mathbf{p}[t], \mathbf{p}[k] \leftarrow \mathbf{p}[k], \mathbf{p}[t]$ # pivot the indices
    $L[\mathbf{p}[k], k] \leftarrow \sqrt{\mathbf{d}[\mathbf{p}[k]]}$
    $\mathbf{l}_{new} \leftarrow K[\mathbf{p}[(k+1):n], \mathbf{p}[k]]$
    $L\left[\mathbf{p}\left[(k+1):n\right], k\right] \leftarrow \frac{1}{L[\mathbf{p}[k],k]} * \left(\mathbf{l}_{new} - L\left[\mathbf{p}\left[(k+1):n\right], 1:(k-1)\right] * L[\mathbf{p}[k], 1:(k-1)]^{\top}\right)$
    $\mathbf{d}[\mathbf{p}[k:n]] \leftarrow \mathbf{d}[\mathbf{p}[k:n]] - (L[\mathbf{p}[k:n], k]).\hat{\ }2$ # end of partial Cholesky part
    **if** $\sigma$ is given **then** # Need to do the augmentation
        $L[n+k, k] \leftarrow \sigma$
    **end if**
    # start of QR part
    $R[1:(k-1), k] \leftarrow Q[:, 1:(k-1)]^{\top} * L[:, k]$
    $Q[:, k] \leftarrow L[:, k] - Q[:, 1:(k-1)] * R[1:(k-1), k]$
    $R[k, k] \leftarrow \|Q[:, k]\|$
    $Q[:, k] \leftarrow Q[:, k]/R[k, k]$
    **return** **p**, L, Q, R, **d**
**end procedure**

After the CholQR_mStep completes, the following post conditions hold true:

(i) $\mathbf{p}[1:m]$ has the same set of elements as $\mathcal{I}_m$;

(ii) $L[\mathbf{p}, 1:m]$ is lower trapezoidal, and it is the rank-$m$ partial Cholesky factor of $K[\mathbf{p}, \mathbf{p}]$;

(iii) $L[\mathbf{p}[1:m], 1:m]$ is lower triangular, and it is the (complete) Cholesky factor of $K[\mathbf{p}[1:m], \mathbf{p}[1:m]]$;

(iv) $\mathbf{d}[\mathbf{p}[1:m]] = 0$ and $\mathbf{d} = diag\left(K - L[1:n, 1:m]L[1:n, 1:m]^{\top}\right)$

(v) if the augmentation trick is required by supplying $\sigma$, then $L[1:m, 1:m] = \sigma I_{m \times m}$, where $I_{m \times m}$ is the rank $m$ identity matrix;

(vi) with or without the augmentation, $L[:, 1:k] = Q[:, 1:k]R[1:k, 1:k]$    $\forall k \in \{1, \ldots, m\}$.

## 2 Efficient Pivot Permutation and Removal

Given $k < m$, the following procedure permute pivot at position $k$ to the right most column of $L$, $Q$, and $R$, after when it would be ready to be removed.

**procedure** CHOLQR_PERMUTETORIGHT(k, m, n, **p**, L, Q, R, **d**, is_augmented)
    **for** $s = k \rightarrow (m-1)$ **do**
        $\mathbf{p}[s], \mathbf{p}[s+1] \leftarrow \mathbf{p}[s+1], \mathbf{p}[s]$ # pivot the indices
        $Q1, R1 \leftarrow qr22(L[\mathbf{p}[s:(s+1)], s:(s+1)]^{\top})$
        $L[\mathbf{p}[s:n], s:(s+1)] \leftarrow L[\mathbf{p}[s:n], s:(s+1)] * Q1$
        $L[\mathbf{p}[s], s+1] \leftarrow 0$
        $R[1:m, s:(s+1)] = R[1:m, s:(s+1)] * Q1$
        $Q2, R2 \leftarrow qr22(R[s:(s+1), s:(s+1)])$
        $R[s:(s+1), 1:m] \leftarrow Q2^{\top} * R[s:(s+1), 1:m]$
        $Q[:, s:(s+1)] \leftarrow Q[:, s:(s+1)] * Q2$
        $R[s+1, s] \leftarrow 0$
        **if** is_augmented **then**
            # rows corresponding to the augmented portion needs not be permuted, handle that
            $Q[n + (s:(s+1)), 1:m] \leftarrow Q1^{\top} * Q[n + (s:(s+1)), 1:m]$
        **end if**
    **end for**
    **return** **p**, L, Q, R, **d**
**end procedure**

where qr22 is a routine to compute the QR factorization of a 2 by 2 matrix. If (i) - (vi) of the previous section hold as pre-conditions for CholQR_PermuteToRight, then they also hold as post-conditions.

Finally, to remove pivot at the last position from the factorization:

**procedure** CHOLQR_REMOVELAST(m, n, $\mathbf{p}$, L, Q, R, $\mathbf{d}$)
    $\mathbf{d}[\mathbf{p}[m:n]] \leftarrow \mathbf{d}[\mathbf{p}[m:n]] + (L[\mathbf{p}[m:n],m]).\hat{}2$
    $L[:,m] \leftarrow 0$
    $Q[:,m] \leftarrow 0$
    $R[1:m,m] \leftarrow 0$
    $R[m,1:m] \leftarrow 0$
    **return** L, Q, R, $\mathbf{d}$
**end procedure**

## 3 Variants

There are two slight variants to the main algorithm as mentioned in the paper, both of which differ in the way information pivots are selected.

The first variant (OI) uses optimized information pivots as in the CSI algorithm instead of randomly chosen ones. More specifically, each time, a new information pivot needs to be selected, we take the one that has the maximum $\mathbf{d}$ values, where $\mathbf{d}$ defined in Sec. 1 of this supplementary material, is the amount of prior variance at that point which is not yet explained by the existing factorization.

The second variant (AA) actively adapt the size of information pivot size. Initialized to a small size, and given an upper bound $z$, this variant exponentially grows the information pivot set size whenever a proposed candidate is rejected, and shrinks it linearly whenever one is accepted. The idea behind the AA variant is the following: as in most optimizations, large progress should be easier to achieve at the beginning comparing to later when closer to convergence, hence less computation is needed to construct a careful approximation at early stages.

## 4 Experimental Details

To ensure fair comparison, all methods discussed in the discrete domain experiments (CholQR, Random, Titsias', and IVM) use the same code for computing the variational free energy objective function and its gradient. For the discrete inducing point selection part of IVM, we use Lawrence's IVM toolbox (from `dcs.shef.ac.uk/people/N.Lawrence/ivm`). All methods except IVM have the same termination criteria: when failed to decrease objective function by a threshold amount, or when exceeded the computational budget. For IVM, because of the inconsistent objectives issue, the objective function values highly fluctuate when alternating between the discrete and continuous phases, as demonstrated in Fig. 3(d) and Fig. 3(3) of the paper. To ensure termination at reasonable time, at any training epoch, we make IVM stop either when there is insufficient change in parameters (no change in inducing points and change in hyperparameters below a predefined threshold), or if the training epoch is larger than 10, and the average relative change in objective function value for the past 10 epochs is below a predefined threshold.

For the bio-informatics dataset, `bindingdb.org` has a large and growing amount of data corresponding to different targets and affinity measures. But at the time of this work, for fixed target and affinity measure, the maximum amount of data we were able to obtain was 3854 sample points (Thrombin as target and dissociation constant ($K_d$) as affinity measure). To run the various sparse GP methods on this dataset, we use a compound kernel consisting of many different labeled and unlabeled graph kernels. Each kernel has its own data variance hyperparameter determining its relevance, which is learned from data during continuous hyper-parameter optimization. The graph kernels are: connected k-node graphlet kernels with 3, 4, and 5 as size of considered graphlets; labeled 3-node graphlet kernel; random walk kernel; labeled random walk kernel; shortest path kernel; labeled graph shortest path kernel; labeled 10-step random walk kernel; 2-step Weisfeiler-Lehman kernel; 2-step Weisfeiler-Lehman edge kernel; 2-step Weisfeiler-Lehman shortest path delta kernel. All the graph kernels are computed using code from `mlcb.is.tuebingen.mpg.de/Mitarbeiter/Nino/Graphkernels`. Additionally, we include two kernels that are redundant but useful to facilitate the automatic relevance learning for all the methods. The first redundant kernel is simply the sum of all graph kernels mentioned above; the second one is a constant identity, which is redundant because GP has a diagonal noise term.

The redundancy makes the optimization less likely to be stuck in bad local minimum. In particular, when trying to automatically learn the relevance of many kernels from data, bad initialization could lead to local minimum where good kernels are dropped while bad ones are kept; this in turn makes it impossible to select good inducing points, causing the optimization to terminate prematurely or to run into numerical issues due to the degeneracy. The two extra redundant kernels allow us to handle this problem without any modification to the learning algorithms or putting explicit prior over hyperparameters. The sum kernel forces all individual kernels to be active at the beginning until the data variance (relevance) on this sum kernel is reduced to zero. In a way, it acts like a temporary weight sharing that is automatically turned off by the optimization once the truly relevant kernels are found. The second constant diagonal kernel reduces the problem of bad local minimum where optimization quickly drives all data variance hyper-parameters to zero, and use very large data noise to explain the observations. This is often the case where bad initial hyper-parameters and/or inducing points give a GP model that cannot interpreting the data at all.

Most important, with the variational energy objective, hyper-parameter optimization for all the methods learned to reduce the data variance (relevance) on these two redundant kernels to zero after a few training epochs; while without this technique, we obtain good solution in most cases, but falls into degeneracy with some initialization.

As for the HoG dataset [1], because there are only three data variance hyperparameters (one for the histogram intersection kernel corresponding to the image from each camera), we did not need to use the redundant kernels in the compound kernel as in BindingDB. We choose to regress from HoG features to the vertical position of the 19 joint markers, leading to 19 different regression problems. In the experiment section of the paper, we present results on one of the 19 problems, results on the other ones are similar and are shown in the next section.

## 5   Addition Experimental Results

In the paper, we show results on one of the 19 regression problems (left wrist), here we present the rest. The setup is exactly the same, with 50 random training/testing splits times 3 random initialization for each regression problem. In each of the figures 1 to 18 that follow, the first two sub-figures show the testing SMSE and SNLP scores averaged over all the 50 random splits and 3 random initialization, shown as function of the number of inducing points. The later two sub-figures show, for one particular run, the testing SMSE and SNLP as function of actual training time. Because the Titsias-16 and Titsias-256 are very slow, we only performed one run for them on each of the regression problem with $m = 32$ inducing points, so we do not have aggregated results across the 50 random splits to show in the first two sub-figures.

Figure 1:

Figure 2:

Figure 3:

Figure 4:

Figure 5:

Figure 6:

Figure 7:

Figure 8:

Figure 9:

Figure 10:

Figure 11:

Figure 12:

Figure 13:

Figure 14:

Figure 15:

Figure 16:

Figure 17:

Figure 18:

## Footnotes

[1] obtained   from   www.maths.lth.se/matematiklth/personal/sminchis/code/TGP. html