[Reviews · NeurIPS 2013]

Submitted by Assigned_Reviewer_7

Update after reading the authors' rebuttal:

Very good paper. I advise the authors, if this paper gets accepted, to update it with the additional information regarding rejection rates and experimental details.

=====
1. Summary

This paper presents an efficient algorithm for optimization of inducing points in sparse Gaussian process regression. While popular approaches such as the IVM use heuristic functions (instead of the marginal likelihood) to do this selection, the proposed method uses the marginal likelihood (or the variational free energy) to optimize inducing points and GP hyper-parameters jointly. Other approaches that achieve this such as the SPGP framework of Snelson or the variational approach by Titisias focus on continuous inputs, while the proposed approach is more suitable for discrete inputs where continuous optimization of inducing points is not feasible.

The paper extends the Cholesky with Side Information (CSI) method of Bach and Jordan to the Gaussian process regression case. In particular, it exploits the observation that it is possible to use the partial Cholesky factorization as a representation of the approximating covariance in the GP. Efficient learning of inducing points is done by updating the corresponding matrices in the factorization and by approximating the reduction in cost of the objective function (marginal likelihood or free energy).

Results are shown on several realistic datasets with suitable comparisons to standard baselines such as IVM, Random and SPGP showing the benefits of their approach, specially in the discrete input case.


2. Quality


The paper is technically sound as it builds upon previous results shown by Bach and Jordan (the CSI method) for kernel methods. The claims are supported by the corresponding derivations of the updates and the approximations required in order to achieve the claimed computational costs. The experimental results also strongly support that the proposed method is faster than other methods such as CSI, the variational framework of Titsias and more effective than other approaches such as random selection and IVM.

3. Clarity


The paper is generally well written providing the detail of the updates necessary for swapping inducing points and the actual optimization algorithm used for the overall optimization framework. However, it would be useful to clarify the following issues:

(a) In section 3.1 there is a small k <= m mentioned in the factored representation (see e.g. line 137) that affects the partial Cholesky factorization and the subsequent QR factorization. However, this k seems to disappear in the following sections. Is this parameter relevant? if So, how is it set in the experiments?


(b) The introduction of the "informative pivots" seems quite crucial in the fast approximate reduction. However, this means that if one were to use this approximation then the method would be optimizing a different objective function. While section 3.4 explains that this fast approximation is only used to consider the candidates for inclusion and that the actual inclusion does evaluate the true changes in the objective function, it is unclear how good this approximation is at selecting good candidates. In other words, what rejection rates are usually observed during optimization?

(c) Figure 1 needs improvement. The axes are not square (with different numerical bounds) and the labels are very small.

(d) More details in the experiments are required. For example, lines 293-294 mention that swapping inducing points at every epoch is unnecessary. How often are the inducing points updated? what about hyper-parameters?

(e) It is unclear why IVM is more expensive than the proposed method as shown in Figure 3 (a). This is probably related to item (d) above.


(f) More analysis in the experiments is needed. Why SPGP is better in terms of predictive likelihood as shown in Figure 5?


(g) (Minor): The end of line 174 may need rephrasing: "same for the updating with j" looks like an incomplete sentence.


4. Originality

The paper addresses the well stablished problem of sparsification of GP regression via inducing points. It mainly extends the results of Bach and Jordan to the sparse Gaussian process regression case. However, it does differ substantially from previous approaches in how it selects these inducing points and how they can be learned efficiently (and jointly with hyper-parameters) while optimizing a single objective function such as the marginal likelihood.

5. Significance

The results are important in that this may establish a superior baseline in terms of inducing selection methods for GP regression. The scalability of such methods is, however, still an issue for these types of algorithms to be used by practitioners on large datasets.
Summary: This paper addresses the well stablished problem of sparsification of GP regression via inducing points. It mainly extends the results of Bach and Jordan to the sparse Gaussian process regression case. However, it does differ substantially from previous approaches in how it selects these inducing points and how they can be learned efficiently (and jointly with hyper-parameters) while optimizing a single objective function such as the marginal likelihood. The experimental evaluation strongly supports the claims in the paper.

Submitted by Assigned_Reviewer_8

The paper proposes an optimisation scheme for sparse GP regression inducing input selection that goes beyond greedy forward selection.

Quality
The optimisation procedure along with the Cholesky based factorisation and the fast approximate scoring heuristic are very reasonable and also the evaluation is of good quality.
i) I'm a bit puzzled about the repeated statement that only a single objective is used to jointly optimise both the inducing points and the continuous hyperparameters (abstract+l.45). As far as I know, Titsias [13] is using the very same single objective and the candidate scoring/greedy growing has the same computational complexity. Also, it is not fully clear when Eq. (4) or Eq. (5) is used in the experiments. Why would one use Eq. (4)?
ii) The statement "code [...] will be made available" at the end of the abstract is somewhat useless for a reviewer since there is no means to very the claim or make sure it becomes true. Adding some demo/code to the supplement would be more convincing.
iii) A weakness of the paper is that is is based on the Nyström or PP covariance matrix approximation by Csató/Opper/Seeger's [3,10]. Using a factorisation similar to the one for sparse EP [a] one could have attempted to use Snelson's FITC or SPGP [12] approximation instead.

Clarity
The paper is well written and the figures are clear.

Originality
Optimising Titsias' [13] variational objective in a non-greedy way is novel as well as the suggested fast approximations for candidate scoring.

Significance
The proposed algorithm and the experimental results suggest that there is a non-negligible benefit in using iterated scoring and replacement instead of greedy growing for inducing point selection in sparse GP regression.

[a] Naish-Guzman and Sean Holden: The Generalized FITC Approximation, NIPS 2007.

Addendum
========
The paper assumes in line 103 that the covariance matrix between the inducing inputs has (full) rank m which is often violated in practice either due to extreme hyperparameters or very similar inducing points. Hence, one needs to add a ridge to stabilise the situation numerically.
Unfortunately, the proposed CholQR algorithm breaks down (without notice) if this strong assumption is not fulfilled and there is no discussion/suggestion in the paper/appendix of how to add a ridge implicitly or to otherwise deal with it in practice.
Also post condition of the appendix (iv) is only correct for the not augmented case and condition (i) is not really correct after calling CholQR_PermuteToRight. It hold only up to index k.
Summary: The proposed optimisation scheme is interesting and the experiments suggest the practical use. However, the paper did consider the less accurate PP [3,10] approximation instead of the SPGP [10] approximation.

Submitted by Assigned_Reviewer_9

Review: Efficient Optimization for Sparse Gaussian Process Regression (NIPS, 586)



Summary of the Paper

The paper proposes a computationally efficient way for selecting a subset of data points for a sparse Gaussian process (GP) approximation. The subset of data points is found by optimizing the same objective as the hyper-parameter learning (e.g., marginal likelihood), and is related to common sparse approximations. The key idea is to exploit efficient updates and downdates of matrix factorization, combined with a computationally efficient approximation of the objective function. The procedure is applicable to both discrete and continuous input domains. The performance analysis shows that the proposed method performs well in both domains, although the gain in a discrete domain is better.





Relation to Previous Work

The proposed method is related to the work by Seeger et al. (2003), i.e., the PP approximation of GPs and the work by Titsias (2009), the variational sparse GPs. Both papers are cited and the paper is put in context of previous work.


Quality

The paper seems technically sound. Its structure is very clear.




Clarity

The paper is written very clearly and very well. The structure is great. What I do like a lot is the next steps are outlined in one sentence at the beginning of a (sub)section. This is really great and makes reading much easier. Because of this property, the paper is a relatively easy read. I'm fully aware that papers like this can be written very differently. Good job!



Originality

The paper presents a novel combination of known algorithms and it performs favorably compared with other sparse GP methods in both discrete and continuous domains.



Significance

The idea of using pseudo inputs, which is used in SPGP (Snelson & Ghahramani), Titsias' sparse method, and also in the GP-LVM (Lawrence), for instance, avoids the selection of a subset of the training data to obtain an approximation of the kernel matrix because of the combinatorial explosion of possible combinations. However, the pseudo-inputs idea does not apply to discrete domains as it is phrased as a continuous optimization problem. The submitted paper instead goes "back" to the idea of selecting a subset of the training data to obtain a sparse GP approximation. This selection is not optimal, but it is clearly better than random. Finding a good subset of the training data can be done computationally quite efficiently. Since this approach is applicable to both discrete and continuous domains, I believe the proposed algorithm can be of some good value to the GP community.




Questions/Comments

1) Could you add a comparison with a full GP as well (all experiments) to have a baseline?

2) Assuming I'm willing to spend O(mn^2) time for the approximation in section 3.3. How much worse does the proposed low-rank approximation with the information pivots perform?



Typos

l.036: myraid ???

l.174: take -> takes

l.431: it's -> its



Summary of the Review:

The paper seems to be solid and is clearly written. I recommend publication.
Summary: A very nice paper with good theoretical results and good experiments.
Author Feedback

Author rebuttal: Thank you for the feedback and questions.

*Assigned_Reviewer_7*
Q: "... a small k <= m ... in the factored representation ... Is this parameter relevant?"

A: k in Sec 3.1 is used solely in the derivation showing that the factorization in Sec. 3.1 applies to any rank 1<=k<=m. It is not a parameter of the algorithm.

Q: "how good this approximation is at selecting good candidates ... what rejection rates ...?"

A: Rejection rates are problem-dependent. They are low in early iterations,
and increase as optimization progresses, as improvement becomes harder.
Let the rejection rate in the 1st j iterations be the number of rejections divided by the number of proposals over j iterations. For the datasets we used, typical rejection rates for j = 200,400,... are
Kin40K: .12, .17, .23, .26, .29, .31, .34
HoG: .20, .29, .35
BindingDB: .04, .13
Rejection rates also tend to increase with decreasing numbers of inducing points, m. With a small inducing set, each point plays a critical role, and is difficult to replace.
Sec. 3.3.2 also explains theoretically why good candidate selection is robust to approximation quality.

Q: "Fig 1 needs improvement"
A: We'll fix it.

Q: " ... How often are inducing points updated? ... hyper-parameters?"

A: For all expts (except 1D example) the number of inducing points swapped per epoch is min(60, m). The max number of function evaluations per epoch in CG hyperparameter optimization is min(20, max(15,2*d)), where d is the number of continuous hyperparameters. Empirically we find that large variation in these limits has little impact on performance (even less than variations in Fig 3c,f).
For more details on the termination criteria, which determine the number of epochs of interleaved discrete and continuous phases, see line 134-142 in the supplementary material.

Q: "why IVM is more expensive ..."

A: Per epoch, IVM is faster, but IVM needs more iterations. If IVM is stopped earlier, its test performance degrades. We have plots showing this, which could be included in supplementary materials.

Q: "Why SPGP is better in terms of predictive likelihood ... in Fig 5?"

A: SPGP is a more flexible model. Continuous relaxation yields more free parameters vs discrete selection, furthermore SPGP can make homoscedastic observation noise goes to zero and use heteroscedastic covariance to model the noise. This leads to better fit on some datasets.

*Assigned_Reviewer_8*
Q: " ... Titsias [13] is using the very same single objective and the candidate scoring/greedy growing has the same computational complexity."

A: In the discrete case, as it was impractical to exhaustively evaluate the true change in objective on all candidates at each step, Titsias[13] proposed to only consider a small subset of candidates. Further, his approach was limited to greedy forward selection. To refine the inducing set after hyperparameter optimization, one needs to drop all previously selected inducing points, and restart from scratch. Unlike a single pass of greedy forward selection, with restarting one cannot guarantee that the variational objective will not increase.
Our formulation allows us to consider all candidates at each step. Also revisiting previous selection is efficient and guaranteed to decrease the objective or else terminate. Empirically, this is significant in both model quality and training time (explained in expt section).
In the continuous case, Titsias[13] did use gradient-based optimization on the same variational objective. We'll revise the paper to clarify this.

Q: "it is not fully clear when Eq. (4) or Eq. (5) is used in the expts. ... "

A: We used (5) on all discrete problems, and for the continuous ones, we showed results with both objectives (4.1, lines 313-315). While our work is about how to optimize either of the objectives, we agree with the arguments in Titsias[13] that (5) is generally superior for sparse GPR.

Q: "... A weakness of the paper is that it is based on the Nyström ... approximation ... . Using a factorisation similar to the one for sparse EP [a] one could ... use Snelson's FITC or SPGP [12] approximation."

A: Using a FITC covariance approximation following [a] would be an interesting extension of our result. We briefly investigated this after deriving the current technique. The extension to the FITC covariance approximation requires a non-trivial derivation. In particular, it is easy to generalize Sec. 3.1 and obtain expressions like (10) with the augmentation trick; then use a QR factorization to arrive at expressions much like (11) and (12). However, Sec 3.2 and 3.3 are more technically challenging to generalize. We are interested in pursuing this in future work.

*Assigned_Reviewer_9*
Q: "... a comparison with a full GP as well ... to have a baseline?"

A: Our baseline is the random subset selection, but we could add comparisons to full GP. As a generic baseline for all sparsification methods, we feel that random selection of subsets of size m, as is done now, is more appropriate, since the corresponding inference time complexity is the same as the sparse GPs with the same number of inducing points - not so with the full GP.

Q: "... willing to spend O(mn^2) time for the approximation ... How much worse does the proposed low-rank approximation with the information pivots perform?"

A: Interesting question, but hard to test except on small datasets (due to long training times). Sec. 3.3.2 gives theoretical intuitions about robustness of the optimization to the quality of the low-rank approximation with info-pivots. Based on preliminary results of several new runs on the BindingDB dataset with O(mn^2) search, it appears that a low-rank approximation with info-pivots is only marginally worse than the full O(mn^2) search (similar to the variations shown in Fig 3c, 3f between CholQR variants and other methods). More comprehensive experiments are needed to confirm this early observation.